

# A combined association of serum uric acid, alanine aminotransferase and waist circumference with non-alcoholic fatty liver disease: a community-based study

Min Wang[1,*], Minxian Wang[1,*], Ru Zhang[1], Liuxin Zhang[2],
Yajie Ding[1], Zongzhe Tang[1], Haozhi Fan[3], Hongliang Wang[4],
Wei Zhang[5], Yue Chen[1] and Jie Wang[1]

[1] School of Nursing, Nanjing Medical University, Nanjing, Jiangsu, China
[2] Affiliated Brain Hospital of Nanjing Medical University, Nanjing, Jiangsu, China
[3] Department of Information, The First Affiliated Hospital of Nanjing Medical University, Nanjing, Jiangsu, China
[4] Mofan West Road Community Health Service Center, Nanjing, Jiangsu, China
[5] Department of Epidemiology, Shanghai Cancer Institute, Shanghai, China
* These authors contributed equally to this work.

Corresponding author
Jie Wang, wangjienjmu@126.com

## ABSTRACT

**Background:** Increasing evidence has supported that serum uric acid (SUA), alanine aminotransferase (ALT) and waist circumference (WC) are associated with the occurrence of non-alcoholic fatty liver disease (NAFLD). However, the combined role of these factors in early screening of NAFLD has not been investigated. We aimed to de lineate this role in a community-based population.

**Methods:** Binary logistic regression was used to explore the correlations of SUA, ALT and WC with NAFLD risk. The goodness of fit and discriminative ability of the model were evaluated by the Hosmer-Lemeshow test and area under the receiver operating characteristic curve (AUROC), respectively.

**Results:** Logistic regression analysis indicated that elevated SUA (adjusted odds ratio (OR) = 2.44, 95% confidence interval (CI) [1.76–3.38]), ALT (adjusted OR = 4.98, 95% CI [3.41–7.27]) and WC (adjusted OR = 3.22, 95% CI [2.01–5.16]) were facilitating factors for incident NAFLD after fully adjusted for related confounders. In addition, the risk of NAFLD followed linear trend s with increasing levels of these three indicators (all $P_{trend} < 0.001$). The risk assessment model consisting of SUA, ALT, WC and demographics showed useful discrimination by AUROC being 0.825 (95% CI [0.811–0.838]) and good performance of calibration ($P = 0.561$).

**Conclusions:** SUA, ALT and WC were all associated with NAFLD, independent of known risk factors. The simple model composed of these indicators showed good performance in the Chinese population, which may be applicable for appraisal of NAFLD risk in primary healthcare.

## INTRODUCTION

Non-alcoholic fatty liver disease (NAFLD), a metabolic syndrome (MetS) manifested in the liver (*Buzzetti, Pinzani & Tsochatzis, 2016*), is pathophysiologically related to adipose tissue inflammation and insulin resistance (*Tilg, Adolph & Moschen, 2021*). It has been widely established that NAFLD has bidirectional association with the components of MetS (obesity, hyperglycemia, hypertension and dyslipidemia), which leads to increased risks of liver-related and extrahepatic complications such as cirrhosis, diabetes (*Hamed et al., 2019*), cerebral stroke (*Abdeldyem et al., 2017*), cancer (*Wang et al., 2020*) and cardiovascular disease (*Targher, Byrne & Tilg, 2020*). NAFLD is currently acknowledged as the leading cause of liver disease worldwide (*Younossi et al., 2016*), and the burden of NAFLD in China has increased considerably due to the changed lifestyle and insufficient diagnostic tools in the past two decades (*Abo-Amer et al., 2020*; *Zhou et al., 2020*). Although liver biopsy is the gold standard for diagnosis of NAFLD, it is costly and impractical for large-scale population (*Chalasani et al., 2018*; *Rinella, 2015*). Thus, readily measurable biochemical and anthropometric markers of NAFLD should be pinpointed, with the purpose of early identification of this disease.

In term of biochemical parameter, serum uric acid (SUA), as the end product of purine metabolism, has been reported to lead endothelial dyfunction, release pro-inflammatory cytokines, induce insulin resistance and hepatocyte fat accumulation (*Sun et al., 2016*; *Zhu et al., 2014*). Large epidemiological studies have demonstrated that increased SUA was correlated with NAFLD risk (*Sirota et al., 2013*; *Wei et al., 2020*; *Zheng et al., 2017*).

In addition, alanine aminotransferase (ALT) and aspartate aminotransferase (AST) are common liver enzymes, which could be released into serum if hepatocellular injury or apoptosis occured. However, ALT is a more liver-specific marker than AST since that AST is widely distributed not only in hepatocytes, but also in skeletal muscle, heart, kidney, and so on (*Collier & Bassendine, 2002*). In fact, ALT has been globally regarded as a reliable biomarker of the liver disease. Studies have suggested that even a mild or no serum ALT elevation can also indicate the existence of liver damage (*Fracanzani et al., 2008*; *Kang et al., 2018*; *Omagari et al., 2011*).

A simple anthropometric parameter, waist circumference (WC) is widely used as an indicator for central obesity (including subcutaneous and visceral fat). Evidence has indicated that NAFLD is primarily caused by visceral fat deposition (*Abdelmoemen et al., 2019*; *Casagrande et al., 2020*). Although having limited precision in differentiating subcutaneous fat from visceral fat, WC has been proven highly correlated with visceral fat (*Al-Radaideh et al., 2019*; *Eloi et al., 2017*). Indeed, insulin resistance and metabolic disorders, the important factors in the pathogenesis of NAFLD, may be partially indicated by elevated WC (*Jiang et al., 2019*).

In spite of the evidence that SUA, ALT and WC all contribute to the onset of NAFLD, using the combination of these three indicators to assess NAFLD risk has rarely been discussed. Therefore, we conducted the community-based study to verify the association of SUA, ALT and WC with NAFLD occurrence, and evaluate the joint validity of them for NAFLD risk assessment.

## MATERIALS AND METHODS

### Design and participants

Participants who underwent physical check-ups in a community in Nanjing (Jiangsu, China) were enrolled in this cross-sectional study from July to September 2018.
The exclusion criteria of our study were as follows: (1) younger than 18 years old; (2) with other liver diseases, such as viral hepatitis, liver cancer and cirrhosis; (3) excessive alcohol intake (≥30 g/d for males and ≥20 g/d for females); (4) with history of liver transplant in the past year, multi-system illness or malignancy; (5) history of psychiatric illness. Diagnosis of NAFLD followed the "*Guideline of prevention and treatment for nonalcoholic fatty liver disease: a 2018 update*" (*National Workshop on Fatty Liver & Alcoholic Liver Disease CSoH, Chinese Medical Association & Fatty Liver Expert Committee CMDA, 2018*), including evidence of hepatic steatosis in images or histology and absence of secondary causes of liver fat accumulation (*e.g.*, heavy alcohol intake, medication and autoimmune hepatitis). With the prevalence of NAFLD in Asia estimated as 30% (*Li et al., 2019*; *Younossi et al., 2018*; *Younossi et al., 2019*), confidence level 1-α set at 0.95 and two-sided confidence interval width set at 0.06, we obtained a minimal sample size of 928 by NCSS-PASS (version 15.0, Dawson edition; Kaysville, UT, USA). In fact, a total of 3,311 participants were recruited in our study, fulfilling the required sample size. To confirm the validity of the study, *post-hoc* power analyses were performed by G*power (version 3.1.9.6), showing that the statistical power of the study exceeded 99%. The study protocol was approved by the Institutional Ethics Review Committee of the Nanjing Medical University (Nanjing, China) with approval number (2018) 596, and all participants provided written informed concents.

### Data collection

Questionnaires and electronic medical record system were used to collect demographic data and medical history. Trained physicians were responsible for the measurement of height, weight, WC and blood pressure. Blood pressure was measured twice with a 5-min interval by a mercury sphygmomanometer and a stethoscope, then averaged for analysis. Body mass index (BMI) was calculated as weight in kilograms divided by the square of height in meters ($kg/m^2$). Abdominal ultrasonography was operated for imaging diagnosis of NAFLD by technicians using a Logiq E9 ultrasound system (General Electric (GE) Healthcare, Milwaukee, WI, USA). Morever, venous blood samples were collected from all participants after an overnight fast for biochemical assessments of ALT, AST, SUA, fasting plasma glucose (FPG), triacylglycerol (TG) and total cholesterol (TC). All the laboratory parameters were analyzed by an automatic biochemical analyzer (Mindray BC-860; Mindray, Shenzhen, China).

### Statistical analysis

Continuous variables were presented as mean ± standard deviation (normal distribution) or median and interquartile range (non-normal distribution); categorical variables were presented as frequencies. Baseline characteristics of subjects between NAFLD and non-NAFLD groups were compared by Chi-square test, independent-samples T test or

Mann-Whitney U test, wherever appropriate. To facilitate the interpretation of associations between SUA, ALT and WC levels with NAFLD risk and screen factors influencing them, all participants were categorized according to SUA quartiles (Q1, ≤280 µmol/L; Q2, 281–337 µmol/L; Q3, 338–394 µmol/L; Q4, ≥395 µmol/L), ALT quartiles (Q1, ≤13 IU/L; Q2, 14–19 IU/L; Q3, 20–29 IU/L; Q4, ≥30 IU/L) and WC quartiles (Q1, ≤75 cm; Q2, 76–82 cm; Q3, 83–89 cm; Q4, ≥90 cm) for further analysis.

The comparisons among quartile groups were conducted by Chi-square test, univariate analysis of variance (ANOVA) or Welch's test. For differences in NAFLD prevalence among quartiles, the Chi-square linear trend test was used. Multivariate logistic regression was employed to investigate the correlations of SUA, ALT and WC with NAFLD risk, and screen predictors to build a model for NAFLD risk assessment. Morever, the assumed linearity between logits of the dependent variable and the continuous predictors was assessed by Box-Tidwell Test (*Thompson, Xie & White, 2003*). Tolerance and variance inflation factor (VIF) were calculated by linear regression analysis and applied for collinearity diagnostics. Discrimination and calibration of the model were analyzed by the area under receiver operating characteristic curve (AUROC) and Hosmer–Lemeshow test, respectively. All the statistical analyses were based on the software SPSS version 23.0 (SPSS Inc., Chicago, IL, USA) and MedCalc version 19.1 (MedCalc Inc., Ostend, Belgium). Two-tailed $P$ values less than 0.05 were considered statistically significant.

## RESULTS

### Baseline characteristics of subjects

Among the 3,311 subjects included in our study, 1,233 (37.2%) had NAFLD. As shown in Table 1, participants with NAFLD were more likely to be male, older and showed significant differences in BMI, WC, SBP, DBP, ALT, AST, SUA, FPG, TG, TC compared with non-NAFLD subjects (all $P < 0.001$).

### Comparison of characteristics among different SUA, ALT and WC groups respectively

After separated by SUA quartiles, significant differences were observed in the distribution of gender, BMI, WC, SBP, DBP, ALT, AST, TG, TC (all $P < 0.001$) and age ($P = 0.030$) among the four groups, but not found in FPG ($P = 0.197$) (Table 2). As shown in Table 3, variations across ALT quartile groups in terms of gender, age, BMI, WC, SBP, DBP, AST, SUA, TG, TC (all $P < 0.001$) and FPG ($P = 0.001$) were statistically significant. Similarly, comparison among the four groups of WC quartiles showed remarkable differences in gender, age, BMI, SBP, DBP, ALT, AST, SUA, FPG, TG, TC (all $P < 0.001$) (Table 4). Moreover, we compared characteristics between high level and normal level groups divided according to the clinical cutoff points for SUA, ALT and WC, whose results further strengthened the reliability of the results above (Tables S1–S3).

As illustrated in Fig.1, the prevalence of NAFLD increased progressively with increasing SUA quartiles (15.7% in Q1, 31.2% in Q2, 42.4% in Q3 and 59.8% in Q4; $P_{trend} < 0.001$), ALT

**Table 1 Baseline characteristics of subjects with and without NAFLD.**

| Variables | non-NAFLD (n = 2,078) | NAFLD (n = 1,233) | t/χ²/Z | P |
|---|---|---|---|---|
| Gender (male), n (%) | 1,379 (66.4) | 1,065 (86.4) | 160.348 | **<0.001**[a] |
| Age (years) | 38.92 ± 9.31 | 42.04 ± 9.29 | −9.331 | **<0.001**[b] |
| BMI (kg/m²) | 22.25 ± 2.45 | 25.43 ± 2.56 | −35.576 | **<0.001**[b] |
| WC (cm) | 79.35 ± 8.44 | 89.03 ± 7.95 | −32.705 | **<0.001**[b] |
| SBP (mmHg) | 122.28 ± 14.11 | 130.99 ± 15.48 | −16.172 | **<0.001**[b] |
| DBP (mmHg) | 73.50 ± 9.51 | 79.75 ± 10.77 | −16.851 | **<0.001**[b] |
| ALT (IU/L)[†] | 16.0 (12.0, 23.0) | 27.0 (19.0, 39.0) | −23.962 | **<0.001**[c] |
| AST (IU/L) | 19.22 ± 7.24 | 22.83 ± 8.51 | −12.453 | **<0.001**[b] |
| SUA (μmol/L) | 317.83 ± 77.93 | 378.18 ± 80.78 | −21.251 | **<0.001**[b] |
| FPG (mmol/L) | 3.44 ± 1.43 | 4.01 ± 1.84 | −9.290 | **<0.001**[b] |
| TG (mmol/L)[†] | 0.97 (0.74, 1.29) | 1.65 (1.19, 2.25) | −27.637 | **<0.001**[c] |
| TC (mmol/L) | 4.48 ± 0.80 | 4.78 ± 0.90 | −9.917 | **<0.001**[b] |

Notes:
BMI, body mass index; WC, waist circumference; SBP, systolic blood pressure; DBP, diastolic blood pressure; ALT, alanine aminotransferase; AST, aspartate aminotransferase; SUA, serum uric acid; FPG, fasting plasma glucose; TG, triacylglycerol; TC, total cholesterol.
[†] Non-normally distributed variables.
[a] Chi square test.
[b] Independent-samples T-test.
[c] Mann-Whitney U-test.
Results in bold type indicate statistically significant.

**Table 2 Comparison of characteristics according to SUA quartiles.**

| Variables | Q1 (n = 829) SUA ≤ 280 | Q2 (n = 827) 281 ≤ SUA ≤ 337 | Q3 (n = 830) 338 ≤ SUA ≤ 394 | Q4 (n = 825) SUA ≥ 395 | P |
|---|---|---|---|---|---|
| Gender (male), n (%) | 250 (30.2) | 618 (74.7) | 768 (92.5) | 808 (97.9) | **<0.001**[a] |
| Age (years)[§] | 40.55 ± 8.36 | 40.57 ± 9.51 | 39.50 ± 9.83 | 39.69 ± 9.89 | **0.030**[c] |
| BMI (kg/m²) | 21.80 ± 2.62 | 22.98 ± 2.69 | 23.93 ± 2.78 | 25.03 ± 2.64 | **<0.001**[b] |
| WC (cm)[§] | 76.19 ± 8.61 | 81.83 ± 8.48 | 85.48 ± 8.35 | 88.52 ± 7.77 | **<0.001**[c] |
| SBP (mmHg) | 119.35 ± 14.90 | 124.63 ± 14.54 | 127.60 ± 14.74 | 130.52 ± 14.47 | **<0.001**[b] |
| DBP (mmHg) | 72.01 ± 9.83 | 75.46 ± 10.32 | 76.58 ± 9.76 | 79.26 ± 10.56 | **<0.001**[b] |
| ALT (IU/L)[†§] | 14.0 (11.0, 19.0) | 18.0 (13.0, 25.0) | 22.0 (16.0, 32.25) | 27.0 (18.0, 40.0) | **<0.001**[c] |
| AST (IU/L)[§] | 17.98 ± 7.46 | 19.41 ± 6.95 | 21.19 ± 7.11 | 23.68 ± 8.89 | **<0.001**[c] |
| FPG (mmol/L) | 3.59 ± 1.65 | 3.75 ± 1.67 | 3.62 ± 1.61 | 3.64 ± 1.54 | 0.197[b] |
| TG (mmol/L)[†§] | 0.91 (0.67, 1.22) | 1.07 (0.81, 1.46) | 1.24 (0.90, 1.78) | 1.57 (1.10, 2.24) | **<0.001**[c] |
| TC (mmol/L) | 4.47 ± 0.83 | 4.53 ± 0.85 | 4.62 ± 0.80 | 4.76 ± 0.88 | **<0.001**[b] |

Notes:
BMI, body mass index; WC, waist circumference; SBP, systolic blood pressure; DBP, diastolic blood pressure; ALT, alanine aminotransferase; AST, aspartate aminotransferase; FPG, fasting plasma glucose; TG, triacylglycerol; TC, total cholesterol.
[†] Non-normally distributed variables after lg transformation.
[§] Variables with heteroscedastic variances among the four groups.
[a] Chi square test.
[b] Univariate analysis of variance (ANOVA).
[c] Welch's test.
Results in bold type indicate statistically significant.

quartiles (12.6% in Q1, 27.0% in Q2, 45.9% in Q3 and 64.5% in Q4; $P_{trend} < 0.001$), and WC quartiles (6.9% in Q1, 23.8% in Q2, 42.9% in Q3 and 71.3% in Q4; $P_{trend} < 0.001$), respectively.

**Table 3  Comparison of characteristics according to ALT quartiles.**

| Variables | Q1 (*n* = 810) ALT ≤ 13 | Q2 (*n* = 884) 14 ≤ ALT ≤ 19 | Q3 (*n* = 809) 20 ≤ ALT ≤ 29 | Q4 (*n* = 808) ALT ≥ 30 | *P* |
|---|---|---|---|---|---|
| Gender (male), *n* (%) | 344 (42.5) | 663 (75.0) | 692 (85.5) | 745 (92.2) | **<0.001**[a] |
| Age (years) | 39.22 ± 9.00 | 40.90 ± 9.52 | 40.68 ± 9.42 | 39.44 ± 9.64 | **<0.001**[b] |
| BMI (kg/m$^2$)[§] | 21.67 ± 2.44 | 22.95 ± 2.64 | 23.97 ± 2.69 | 25.19 ± 2.75 | **<0.001**[c] |
| WC (cm) | 76.22 ± 8.17 | 82.01 ± 8.43 | 85.00 ± 8.46 | 88.83 ± 8.28 | **<0.001**[b] |
| SBP (mmHg) | 120.08 ± 14.46 | 124.17 ± 14.34 | 127.01 ± 15.20 | 130.96 ± 14.88 | **<0.001**[b] |
| DBP (mmHg)[§] | 72.27 ± 9.41 | 74.85 ± 9.90 | 76.66 ± 10.75 | 79.62 ± 10.35 | **<0.001**[c] |
| AST (IU/L)[§] | 15.23 ± 2.42 | 17.62 ± 2.88 | 20.52 ± 3.61 | 29.17 ± 10.85 | **<0.001**[c] |
| SUA (μmol/L) | 291.06 ± 72.52 | 327.73 ± 78.07 | 356.02 ± 75.27 | 387.70 ± 79.80 | **<0.001**[b] |
| FPG (mmol/L)[§] | 3.48 ± 1.48 | 3.62 ± 1.50 | 3.73 ± 1.59 | 3.77 ± 1.88 | **0.001**[c] |
| TG (mmol/L)[†§] | 0.88 (0.67, 1.16) | 1.07 (0.80, 1.44) | 1.27 (0.92, 1.92) | 1.59 (1.13, 2.25) | **<0.001**[c] |
| TC (mmol/L) | 4.41 ± 0.81 | 4.54 ± 0.80 | 4.60 ± 0.82 | 4.83 ± 0.92 | **<0.001**[b] |

Notes:

BMI, body mass index; WC, waist circumference; SBP, systolic blood pressure; DBP, diastolic blood pressure; AST, aspartate aminotransferase; SUA, serum uric acid; FPG, fasting plasma glucose; TG, triacylglycerol; TC total cholesterol.
[†] Non-normally distributed variables after lg transformation.
[§] Variables with heteroscedastic variances among the four groups.
[a] Chi square test.
[b] Univariate analysis of variance (ANOVA).
[c] Welch's test.
Results in bold type indicate statistically significant.

**Table 4  Comparison of characteristics according to WC quartiles.**

| Variables | Q1 (*n* = 725) WC ≤ 75 | Q2 (*n* = 780) 76 ≤ WC ≤ 82 | Q3 (*n* = 905) 83 ≤ WC ≤ 89 | Q4 (*n* = 818) WC ≥ 90 | *P* |
|---|---|---|---|---|---|
| Gender (male), *n* (%) | 212 (29.2) | 546 (70.0) | 825 (91.2) | 790 (96.6) | **<0.001**[a] |
| Age (years)[§] | 38.51 ± 8.61 | 39.69 ± 9.27 | 40.57 ± 9.27 | 42.48 ± 9.58 | **<0.001**[b] |
| BMI (kg/m$^2$)[§] | 20.37 ± 1.83 | 22.38 ± 1.78 | 24.05 ± 1.76 | 26.52 ± 2.29 | **<0.001**[b] |
| SBP (mmHg)[§] | 117.25 ± 14.46 | 122.83 ± 13.39 | 127.31 ± 13.51 | 133.36 ± 15.25 | **<0.001**[b] |
| DBP (mmHg)[§] | 70.98 ± 9.39 | 73.90 ± 9.37 | 76.94 ± 9.95 | 81.05 ± 10.37 | **<0.001**[b] |
| ALT (IU/L)[†§] | 13.0 (10.0, 18.0) | 16.5 (13.0, 23.0) | 22.0 (16.0, 31.0) | 27.5 (19.0, 41.0) | **<0.001**[b] |
| AST (IU/L)[§] | 18.02 ± 6.04 | 19.15 ± 6.78 | 21.06 ± 8.29 | 23.40 ± 8.66 | **<0.001**[b] |
| SUA (μmol/L)[§] | 275.00 ± 65.23 | 326.78 ± 73.56 | 357.80 ± 73.96 | 389.54 ± 79.64 | **<0.001**[b] |
| FPG (mmol/L)[§] | 3.28 ± 1.29 | 3.56 ± 1.46 | 3.74 ± 1.60 | 4.09 ± 1.94 | **<0.001**[b] |
| TG (mmol/L)[†§] | 0.83 (0.64, 1.11) | 1.05 (0.80, 1.45) | 1.27 (0.94, 1.79) | 1.65 (1.15, 2.26) | **<0.001**[b] |
| TC (mmol/L)[§] | 4.42 ± 0.79 | 4.54 ± 0.81 | 4.61 ± 0.81 | 4.80 ± 0.96 | **<0.001**[b] |

Notes:

BMI, body mass index; WC, waist circumference; SBP, systolic blood pressure; DBP, diastolic blood pressure; ALT, alanine aminotransferase; AST, aspartate aminotransferase; SUA, serum uric acid; FPG, fasting plasma glucose; TG, triacylglycerol; TC, total cholesterol.
[†] Non-normally distributed variables after lg transformation.
[§] Variables with heteroscedastic variances among the four groups.
[a] Chi square test.
[b] Welch's test.
Results in bold type indicate statistically significant.

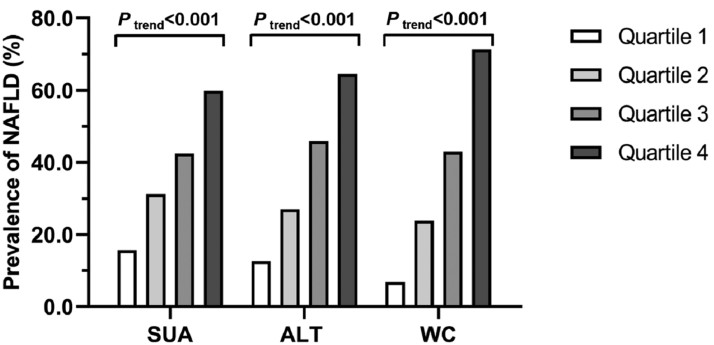

**Figure 1 Prevalence of NAFLD in quartiles of SUA, ALT and WC, respectively.** SUA: Q1, ≤280 μmol/L; Q2, 281–337 μmol/L; Q3, 338–394 μmol/L; Q4, ≥395 mmol/L. ALT: Q1, ≤13 IU/L; Q2, 14–19 IU/L; Q3, 20–29 IU/L; Q4, ≥30 IU/L. WC: Q1, ≤75 cm; Q2, 76–82 cm; Q3, 83–89 cm; Q4, ≥90 cm.

## Associations of SUA, ALT and WC with NAFLD risk respectively

The associations of SUA, ALT and WC with NAFLD risk were assessed through multivariate logistic regression (Table 5). Considering that gender, age, BMI, SBP, DBP, AST, FPG, TG and TC were all significantly different among quartile groups of SUA, ALT or WC, we adjusted them as covariates for further analyses.

SUA Q2 (crude OR = 2.44, 95% CI [1.92–3.09]), Q3 (crude OR = 3.96, 95% CI [3.14–5.00]), and Q4 (crude OR = 7.98, 95% CI [6.32–10.08]) were associated with higher odds of NAFLD compared to Q1, and this association was dose-dependent ($P_{trend}$ < 0.001). After adjustment for gender, age, BMI, SBP, DBP (model 2), the NAFLD risk increased significantly with the increases of SUA quartiles ($P_{trend}$ < 0.001). Similar associations between SUA levels and NAFLD risk were obtained after further adjusted for AST, FPG, TG and TC (Q2 *vs* Q1: adjusted OR = 1.66, 95% CI [1.23–2.22]; Q3 *vs* Q1: adjusted OR = 1.92, 95% CI [1.40–2.63]; Q4 *vs* Q1: adjusted OR = 2.44, 95% CI [1.76–3.38]; $P_{trend}$ < 0.001).

Positive associations existed between ALT quartiles and NAFLD risk, with ORs significantly increasing from 2.57 (95% CI [1.99–3.32]) to 12.60 (95% CI [9.79–16.22]) as the ALT quartiles elevated from Q2 to Q4 (Q1 served as reference) (model 1). Subjects in ALT Q2 (adjusted OR = 1.62, 95% CI [1.21–2.17]), Q3 (adjusted OR = 3.03, 95% CI [2.26–4.05]) and Q4 (adjusted OR = 4.83, 95% CI [3.57–6.52]) had significantly increased risk of NAFLD compared to Q1 after adjusted for gender, age, BMI, SBP and DBP ($P_{trend}$ < 0.001). Further adjustment for other covariates did not change the positive associations and linear trend (model 3).

Significant correlations between WC quartiles and NAFLD were observed without adjustment of factors (model 1). When adjusting for gender, age, BMI, SBP and DBP, the ORs of NAFLD for WC Q2, Q3, Q4 compared to Q1 were 2.21 (95% CI [1.53–3.20]), 3.00 (95% CI [2.01–4.48]) and 4.31 (95% CI [2.73–6.81]) respectively, which increased progressively with the elevation of WC levels ($P_{trend}$ < 0.001). Progressive increases in the odds of NAFLD with elevated WC quartiles persisted after further adjusted for AST, FPG, TG and TC ($P_{trend}$ < 0.001) (model 3).

**Table 5 Associations of SUA, ALT and WC levels with the prevalence of NAFLD.**

| Quartiles | Model 1 | | | Model 2 | | | Model 3 | | |
|---|---|---|---|---|---|---|---|---|---|
| | OR | 95% CI | P | OR | 95% CI | P | OR | 95% CI | P |
| SUA | | | | | | | | | |
| Q1 | 1 | | | 1 | | | 1 | | |
| Q2 | 2.44 | [1.92–3.09] | <0.001 | 1.81 | [1.36–2.42] | <0.001 | 1.66 | [1.23–2.22] | 0.001 |
| Q3 | 3.96 | [3.14–5.00] | <0.001 | 2.39 | [1.77–3.23] | <0.001 | 1.92 | [1.40–2.63] | <0.001 |
| Q4 | 7.98 | [6.32–10.08] | <0.001 | 3.46 | [2.54–4.73] | <0.001 | 2.44 | [1.76–3.38] | <0.001 |
| $P_{trend}$ | | | <0.001* | | | <0.001* | | | <0.001* |
| ALT | | | | | | | | | |
| Q1 | 1 | | | 1 | | | 1 | | |
| Q2 | 2.57 | [1.99–3.32] | <0.001 | 1.62 | [1.21–2.17] | 0.001 | 1.66 | [1.23–2.24] | 0.001 |
| Q3 | 5.88 | [4.58–7.55] | <0.001 | 3.03 | [2.26–4.05] | <0.001 | 2.84 | [2.08–3.88] | <0.001 |
| Q4 | 12.60 | [9.79–16.22] | <0.001 | 4.83 | [3.57–6.52] | <0.001 | 4.98 | [3.41–7.27] | <0.001 |
| $P_{trend}$ | | | <0.001* | | | <0.001* | | | <0.001* |
| WC | | | | | | | | | |
| Q1 | 1 | | | 1 | | | 1 | | |
| Q2 | 4.23 | [3.04–5.89] | <0.001 | 2.21 | [1.53–3.20] | <0.001 | 2.01 | [1.38–2.93] | <0.001 |
| Q3 | 10.13 | [7.39–13.90] | <0.001 | 3.00 | [2.01–4.48] | <0.001 | 2.45 | [1.62–3.70] | <0.001 |
| Q4 | 33.49 | [24.21–46.34] | <0.001 | 4.31 | [2.73–6.81] | <0.001 | 3.22 | [2.01–5.16] | <0.001 |
| $P_{trend}$ | | | <0.001* | | | <0.001* | | | <0.001* |

Notes:
SUA, serum uric acid; ALT, alanine aminotransferase; WC, waist circumference; OR, odds ratio; CI, confidence interval.
For SUA: Q1, ≤280 μmol/L; Q2, 281–337 μmol/L; Q3, 338–94 μmol/L; Q4, ≥395 μmol/L.
For ALT: Q1, ≤13 IU/L; Q2, 14–19 IU/L; Q3, 20–29 IU/L; Q4, ≥30 IU/L.
For WC: Q1, ≤75 cm; Q2, 76–82 cm; Q3, 83–89 cm; Q4, ≥90 cm.
Model 1: unadjusted.
Model 2: adjusted for gender, age, BMI, SBP, DBP.
Model 3: adjusted for gender, age, BMI, SBP, DBP, AST, FPG, TG, TC.
* P for trend.
Results in bold type indicate statistically significant.

## Validity of SUA, ALT and WC in NAFLD risk assessment

To further verify whether SUA, ALT and WC were independent risk factors of NAFLD or not, we constructed binary logistic regression (forward:LR) with the three variables and demographics (gender and age) included in the continuous form. The Box-Tidwell Test suggested that linearity assumption was met in SUA, WC and age but not in ALT, we therefore transformed ALT into the categorical variable. No collinearity among the included variables was observed in collinearity diagnostics (tolerance > 0.1; VIF < 10). As shown in Table 6, female (OR = 2.355, 95% CI [1.799–3.084]), age (OR = 1.033, 95% CI [1.023–1.042]), ALT > 40 (OR = 2.338, 95% CI [1.791–3.052]), SUA (OR = 1.006, 95% CI [1.005–1.008]) and WC (OR = 1.140, 95% CI [1.125–1.156]) all played significant roles in facilitating the occurrence of NAFLD.

Moreover, a model was developed including these identified predictors, and was calculated as: (−16.229) + 0.857 × Gender (male = 1, female = 2) + 0.032 × Age + 0.849 × ALT (ALT = 1 if > 40, 0 if ≤ 40 IU/L) + 0.006 × SUA + 0.131 × WC. Insignificant result in Hosmer-Lemeshow test of the model represented agreement between predictive

Table 6 **Logistic regression for NAFLD risk assessment based on ALT, SUA, WC and demographics.**

| Variables | Unstandardized β | OR | 95% CI | P |
|---|---|---|---|---|
| Gender (female *vs* male) | 0.857 | 2.355 | [1.799–3.084] | **<0.001** |
| Age | 0.032 | 1.033 | [1.023–1.042] | **<0.001** |
| ALT (>40 *vs* ≤40) | 0.849 | 2.338 | [1.791–3.052] | **<0.001** |
| SUA | 0.006 | 1.006 | [1.005–1.008] | **<0.001** |
| WC | 0.131 | 1.140 | [1.125–1.156] | **<0.001** |
| Constant | −16.229 | – | – | – |
| Hosmer-Lemeshow test: $\chi^2$ = 6.779, $P^*$ = 0.561 | | | | |

**Notes:**
ALT, alanine aminotransferase; SUA, serum uric acid; WC, waist circumference; OR, odds ratio; CI, confidence interval.
* $P$ for Hosmer-Lemeshow test.
Results in bold type indicate statistically significant.

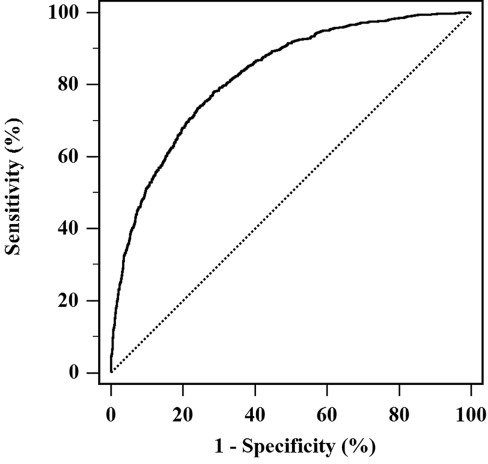

**Figure 2 AUROC of the model for NAFLD risk assessment.** The model developed with NAFLD occurrence as a response variable, and combination of ALT, SUA, WC, age and gender as independent variables. The AUROC of the model was 0.825 (95% CI [0.811–0.838]).

and actual NAFLD probability ($P$ = 0.561). The model had good discrimination capability for NAFLD risk, with AUROC being 0.825 (95% CI [0.811–0.838]) (Fig. 2).

## DISCUSSION

The present study evidenced strong and positive associations between SUA, ALT, WC and NAFLD occurrence, irrespective of multiple metabolic factors. The prevalence of NAFLD increased with the increasing levels of SUA, ALT and WC. Moreover, a simple model composed of these three indicators and demographics (gender and age) showed good capability in NAFLD risk assessment.

Participants with NAFLD accouted for a large proportion (37.2%) in our study, roughly paralleling with the global high prevalence of it (*Younossi et al., 2018*). The huge economic and health burdens carried by the epidemic prompt strong interest in identification of simple, readily measurable biomarkers for early detection of NAFLD. Consequently, we

pooled common biochemical and anthropometric parameters (SUA, ALT, WC) into this study to seek targets of screening strategy.

The independent correlation between serum SUA levels and NAFLD risk has been highlighted by multiple studies (*Liu et al., 2015*; *Shih et al., 2015*; *Sirota et al., 2013*; *Wei et al., 2020*; *Zhou, Wei & Fan, 2016*), which is in accordance with the findings of our study. The pathogenic mechanisms responsible for the association between SUA and NAFLD are essential. *Wan et al. (2016)* have proposed that uric acid may induce liver steatosis and insulin resistance, mediated by the activation of NOD-like receptor family pyrin domain containing 3 (NLRP3) inflammasome. The activation of NLRP3 has been identified essential for the development of liver fibrosis (*Wree et al., 2014*). Morever, SUA could induce fat accumulation in liver through hepatic endoplasmic reticulum stress (*Choi et al., 2014*) and aldose reductase activation (*Sanchez-Lozada et al., 2019*). SUA reduction has been suggested as a new therapeutic target for NAFLD, though more clinical data are needed to determine its effectiveness (*Paschos et al., 2018*; *Sun et al., 2016*).

A positive relationship between ALT levels (even in a normal range) and NAFLD risk was observed in our study, demonstrating that ALT is an independent risk factor for the occurrence of NAFLD. Morever, some previous studies taking ALT as a serum marker for predicting NAFLD have suggested ALT could be applied to the detection of hepatic diseases (*Omagari et al., 2011*), but others reported ALT as poor predictor (*Verma et al., 2013*). The inconsistent results implicate that the predictive reliability of ALT might be further strengthened by combining with multi-indicators.

It has been recognized that abdominal obesity, manifested by WC, causes more obesity-related health risks than total adiposity evaluated by BMI (*Tian et al., 2020*; *Walls et al., 2011*). Obesity-related diseases, such as metabolic syndrome, insulin resistance, cardiovascular disorders, and NAFLD, are all reported to be associated with WC levels (*Dogan et al., 2019*; *Ge et al., 2021*; *Jiang et al., 2019*). Liver has a central role in lipid metabolism, and free fatty acids generated from the huge amount of visceral adiposity tissue lead to the accumulation of lipid in the liver (*Milic, Lulic & Stimac, 2014*). However, there is no approved medication for NAFLD treatment currently, and the most effective treatment is lifestyle modifications including weight loss and diet regulation (*Younossi et al., 2019*). The dose-response relationship between WC levels and NAFLD risk in our study may provide a clue for candidate therapeutic targets.

Results in our study indicated that SUA, ALT and WC were all independent risk factors, which could be potentially modified through behaviour modification for NAFLD treatment and prevention (*Romero-Gomez, Zelber-Sagi & Trenell, 2017*). It has been well acknowledged that gender and age appear to be predictive for NAFLD risk. We further integrated a combination of SUA, ALT, WC, gender and age into a model, which exhibited high prediction ability with useful discrimination and excellent calibration towards NAFLD occurrence. Recent studies (*Hanafy et al., 2019*; *Klisic, Kavaric & Ninic, 2019*) also reported that the combination of ALT and SUA together with many complicated parameters had a higher discrimination to fatty liver. Of note, our model was based on easily accessible biochemical and anthropometric predictors, which may be applicable for screening in primary healthcare settings.

Our study has some limitations. First, individuals with NAFLD in our study were imaging diagnosed by ultrasonography, which is not the gold standard for diagnosis of NAFLD. Though used in more than 99% diagnosis of NAFLD cases in China (*Zhou et al., 2020*), ultrasonography could not provide data about hepatic inflammation or fibrosis. Second, WC is not a good proxy for abdominal visceral fat, and better surrogates for visceral fat accumulation should be included in the future. Third, the cross-sectional study failed to evaluate the changes of relationship between the three predictors and NAFLD risk over time. Fourth, there may be problems in extrapolating the model developed in our study due to the lack of external verification. Hence, we will conduct further studies to verify risk assessment performance of this noninvasive model in other populations.

## CONCLUSIONS

This study found that elevated SUA, ALT and WC were all independent risk factors of NAFLD. A combination of these three indicators and demographics constituted a easy-to-use and efficient model in primary healthcare, which may provide a new approach for early risk appraisal of NAFLD in the Chinese population.

## ACKNOWLEDGEMENTS

We thank Dr. Yongke Cao for providing English langauage polishing support.

### Funding

This work was supported by the Natural Science Foundation of Jiangsu Province (grant number BK20181369), the Six Talent Peak Project in Jiangsu Province (grant number 2019-WSN-049), and the Priority Academic Program Development of Jiangsu Higher Education Institutions (grant number PAPD [2018] 87). The funders had no role in study design, data collection and analysis, decision to publish, or preparation of the manuscript.

### Grant Disclosures

The following grant information was disclosed by the authors:
Natural Science Foundation of Jiangsu Province: BK20181369.
Six Talent Peak Project in Jiangsu Province: 2019-WSN-049.
Jiangsu Higher Education Institutions: PAPD [2018] 87.

### Competing Interests

The authors declare that they have no competing interests.

### Author Contributions

- Min Wang conceived and designed the experiments, performed the experiments, analyzed the data, prepared figures and/or tables, authored or reviewed drafts of the paper, and approved the final draft.
- Minxian Wang performed the experiments, prepared figures and/or tables, and approved the final draft.

- Ru Zhang conceived and designed the experiments, analyzed the data, authored or reviewed drafts of the paper, and approved the final draft.
- Liuxin Zhang performed the experiments, prepared figures and/or tables, and approved the final draft.
- Yajie Ding performed the experiments, prepared figures and/or tables, and approved the final draft.
- Zongzhe Tang performed the experiments, prepared figures and/or tables, and approved the final draft.
- Haozhi Fan conceived and designed the experiments, authored or reviewed drafts of the paper, and approved the final draft.
- Hongliang Wang conceived and designed the experiments, authored or reviewed drafts of the paper, and approved the final draft.
- Wei Zhang conceived and designed the experiments, authored or reviewed drafts of the paper, and approved the final draft.
- Yue Chen conceived and designed the experiments, authored or reviewed drafts of the paper, and approved the final draft.
- Jie Wang conceived and designed the experiments, authored or reviewed drafts of the paper, and approved the final draft.

## Human Ethics

The following information was supplied relating to ethical approvals (*i.e.*, approving body and any reference numbers):

The study protocol was approved by the Institutional Ethics Review Committee of the Nanjing Medical University [Ethics Approval Number (2018) 596].

## Data Availability

The raw measurements are available in the Supplemental File.

## Supplemental Information

Supplemental information for this article can be found online at http://dx.doi.org/10.7717/peerj.13022#supplemental-information.

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
