# Peer review of "A combined association of serum uric acid, alanine aminotransferase and waist circumference with non-alcoholic fatty liver disease: a community-based study"

_PeerJ, doi:10.7717/peerj.13022_

## Round 0.1 · original submission · Major Revisions

After review of the manuscript by the external reviewers and myself, we believe that there are areas for improvement in the manuscript. The authors should respond and carefully address the comments made by the reviewers.

Reviewer 1 ·

Basic reporting

The authors investigated the association of serum uric acid, alanine aminotransferase and waist circumference with non-alcoholic fatty liver disease; this is a hot topic and the study is interesting; However; only minor comments:
=Introduction: You should illustrate more the importance of NAFLD; its association with metabolic syndrome; and extrahepatic manifestations as cerebral stroke or cardiovascular events.
=Methods: You should clarify more the sample size calculation; and the power of the study to confirm the validity of the study.
=Results: Well presented.
=Introduction and discussion: should be enriched with recent studies in the field.
= References: some are old even returning to 1960s; So, I suggest adding more recent and relevant references:
-Diabetes Association with Liver Diseases: An Overview for Clinicians. Endocr Metab Immune Disord Drug Targets. 2019;19(3):274-280. doi: 10.2174/1871530318666181116111945.
-Overexpression of Hepassocin in Diabetic Patients with Nonalcoholic Fatty Liver Disease May Facilitate Increased Hepatic Lipid Accumulation. Endocr Metab Immune Disord Drug Targets. 2019;19(2):185-188.
-Efficacy of a non-invasive model in predicting the cardiovascular morbidity and histological severity in non-alcoholic fatty liver disease. Diabetes Metab Syndr. 2019 May-Jun;13(3):2272-2278.
-Nonalcoholic fatty liver disease in patients with acute ischemic stroke is associated with more severe stroke and worse outcome. J Clin Lipidol. 2017 Jul-Aug;11(4):915-919.
-Relationship Between Helicobacter pylori Infection and Nonalcoholic Fatty Liver Disease (NAFLD) in a Developing Country: A Cross-Sectional Study. Diabetes Metab Syndr Obes. 2020 Mar 2;13:619-625

Experimental design

Research question is well defined, relevant & meaningful. The research fills an identified knowledge gap.

Validity of the findings

The findings are valid. So, I suggest accepting the article for publication.

Reviewer 2 ·

Basic reporting

1. Your introduction needs more detail. I suggest that you improve the description at lines 69-73 to provide more justification for your study. The authors should explain more, why to use WC, when the limitation of WC is inaccurate distinction between visceral and subcutaneous adipose tissue in the abdominal region.

2. As anthropometric parameter, waist circumference (WC) is usually treated as an indicator for central obesity…because they authors not use the Visceral Adiposity Index (VAI), an indicator of adipose distribution and function.

Experimental design

1. “SUA, ALT and WC were classified into categories defined by quartiles.”: The authors need to provide the rationale of conversion of SUA, ALT and WC from a continuous variable into quartiles.
2. Tables 2 -4: The authors should show Comparison of characteristics according to the cutoff points for SUA, ALT and WC (dichotomic variable).
3. Statistical analysis, described with sufficient detail.

Validity of the findings

The absence of a replication group makes the conclusions be taken with caution. Lines 252-255. I suggest softening the conclusions, and indicate…. in the Chinese population.

Additional comments

no comment

Reviewer 3 ·

Basic reporting

The Authors of this manuscript aimed to determine the combined effects of serum uric acid (SUA), alanine aminotransferase (ALT), and waist circumference (WC) on early screening of non-alcoholic fatty liver disease (NAFLD) in a community-based Chinese population. The manuscript is written clearly. The data justify the conclusions. Only some minor changes are recommended.

Experimental design

The Authors should describe in the Methods section how blood pressure was measured.
The units in Table 6 are redundant. Also, the units for SUA are not correct.

Validity of the findings

'no comment'

Additional comments

English editing for some minor corrections is needed.

---

## Round 0.2 · accepted · Accept

The Authors have made corrections according to the Reviewers suggestions.

Reviewer 1 ·

Basic reporting

The authors investigated a combined association of serum uric acid, alanine aminotransferase and waist circumference with non-alcoholic fatty liver disease in a community-based study ; the study is interesting and discusses a hot topic. The authors addressed the comments of the reviewers.So, I suggest acceptance of the article.

Experimental design

Methods were described with sufficient detail and information.

Validity of the findings

Conclusions are well stated and are linked to the original research.

Reviewer 2 ·

Basic reporting

no comment

Experimental design

no comment

Validity of the findings

no comment

Additional comments

no comment

Reviewer 3 ·

Basic reporting

The Authors have made corrections according to the Reviewer's suggestions.

Experimental design

The Authors have made corrections according to the Reviewer's suggestions.

Validity of the findings

The Authors have made corrections according to the Reviewer's suggestions.

Additional comments

No additional comments.